# Advancing Motivation Feedforward Control of Permanent Magnetic Linear Oscillating Synchronous Motor for High Tracking Precision

**Zongxia Jiao** [1,2,3], **Yuan Cao** [1], **Liang Yan** [1,2,3,*], **Xinglu Li** [1,3], **Lu Zhang** [1,2,3] and **Yang Li** [1,3]

[1] School of Automation Science and Electrical Engineering, Beihang University, Beijing 100191, China; zxjiao@buaa.edu.cn (Z.J.); 18103343996@163.com (Y.C.); lixinglu@buaa.edu.cn (X.L.); zhanglu@buaa.edu.cn (L.Z.); leeyoung303@163.com (Y.L.)
[2] Ningbo Institute of Technology, Beihang University, Ningbo 315800, China
[3] Science and Technology on Aircraft Control Laboratory, Beihang University, Beijing 100191, China
* Correspondence: lyan1991@gmail.com

**Abstract:** Linear motors have promising application to industrial manufacture because of their direct motion and thrust output. A permanent magnetic linear oscillating synchronous motor (PMLOSM) provides reciprocating motion which can drive a piston pump directly having advantages of high frequency, high reliability, and easy commercial manufacture. Hence, researching the tracking performance of PMLOSM is of great importance to realizing its popularization and application. Traditional PI control cannot fulfill the requirement of high tracking precision, and PMLOSM performance has high phase lag because of high control stiffness. In this paper, an advancing motivation feedforward control (AMFC), which is a combination of advancing motivation signal and PI control signal, is proposed to obtain high tracking precision of PMLOSM. The PMLOSM inserted with AMFC can provide accurate trajectory tracking at a high frequency. Compared with single PI control, AMFC can reduce the phase lag from −18 to −2.7 degrees, which shows great promotion of the tracking precision of PMLOSM. In addition, AMFC will promote the application of PMLOSM to other working conditions needing high frequency reciprocating tracking performance and give PMLOSM greater future prospects.

**Keywords:** PI control; advancing motivation feedforward control; high frequency and permanent magnetic linear oscillating motor

## 1. Introduction

Linear motors can provide direct motion without any other transmission gears or power producers such as fluid pumps and air compressors, so the linear motor has been vitally applied to rail transit [1], flexible beam systems [2], biaxial systems [3], maglev transportation systems [4], printer gantry positioning [5], H-shape gantry control [6], and other applications that require fast smooth operation [7] or high frequency response [8]. The PMLOSM is a permanent magnetic linear synchronous motor that has been inserted in high stiffness springs to make up the mover as a typical secondary order mass-spring system and can provide high frequency reciprocating motion, so PMLOSMs can be a core component for a directly driven linear hydraulic pump with enough power density [9].

In recent years, researchers have centered on the linear motor performance of faster transient response [10–12], tracking accuracy [13–15], and high robustness [16–19]. The performance of linear motors is influenced by many factors such as parameter uncertainties [11,20], nonlinear friction, kinematic and dynamic constraints [14], nonlinear electromagnetic field [21], exogenous disturbance [22], cogging force [10], and back electromagnetic force [23]. Various approaches are related to linear motor control such as multiple input–output control [10], network control [11,19], speed estimation [12], sliding mode

control [15,16,21,24–26], iterative learning control [17], adaptive robust control [10], sensorless control [23,27,28], backstepping control [29], and impedance control [30,31]. A novel multiple-input multiple-output space-state control model improves direct thrust force control leading to faster transient response of permanent magnetic linear synchronous motor [10]. A radial basis function network control satisfies good transient performance of the linear motor through lumping uncertainties including parameter variations, external disturbances and nonlinear friction force at tracking frequency of 0.25 Hz [11]. A sliding mode controller, injected with estimated speed, is constructed to ensure the accuracy and robustness for linear permanent magnetic synchronous motor (PMLSM) [16], and an adaptive recursive sliding mode control conquers deterioration of parameter uncertainty, nonlinear friction, and exogenous disturbance for the linear motor to get high speed and high precision performance at about 1 Hz [21]. An adaptive robust controller, taking into account nonlinear electromagnetic field, is designed to obtain accurate trajectory performance at 1 Hz [21]. An adaptive backstepping control strategy is provided to obtain satisfied performance of position tracking ability and control robustness for PMLSM [29]. Sensorless control is conducted on linear motor via extended Kalman filter [27,28]. A typical PID controller is designed for the asymmetric bilateral linear hybrid switched reluctance motor to obtain high-accuracy position performance at 0.5 Hz [32].

All these linear motor control studies cannot match PMLOSM it its high tracking precision because of high stiffness, high operating frequency, and low sampling times. High stiffness springs are inserted in the mover of the PMLOSM to give a mass-spring second order system so that the mover can oscillate as a trigonometric waveform at high frequency, which undoubtedly challenges the phase response of PMLOSM. Fortunately, researchers have explored some control strategies for PMLOSM. Yang proposed a dual feedback and feedforward controller for stroke and phase compensation [30] indicating that stroke and phase should each be taken into consideration for high tracking precision of PMLOSM. Du proposed a model-based feedforward controller that practically improved the phase response of linear oscillating motor [31]. Wang introduced B-spline neural network compensator as feedforward control to obtain better performance, but required more conditions to satisfy learning process of feedforward coefficient [33]. These studies indicate that feedforward control can improve phase response. For stroke control, traditional PI control of PMLOSM has the advantages of convenient adjustment and robustness, but cannot cope with phase delay [34]. Though Kim studied PID control on PMLOSM on improving phase response, noise and disturbance of the derivative element led to insufficient control performance [35]. Input and output signals should be taken into consideration for a more efficient controller [36]. Integral action and anticipatory action are taken into consideration mainly in controller design for linear motor position control [37]. Above all, we combined traditional PI control and feedforward control as a two-degree-of-freedom controller [38], to obtain high tracking precision of PMLOSM.

In this paper, we provide an advancing motivation feedforward control (AMFC) strategy for PMLOSM. In Section 2, the structure and problem illustration are given. In Section 3, the mathematical model and simulation analysis are given. In Section 4, the experiment results are given. In Section 5, conclusion is given.

## 2. Structure and Problem Illustration

The PMLOSM can provide reciprocating motion at a high frequency because of an inserted hard spring into the mover to construct a second order mass-spring system. The cross section of the PMLOSM is shown in Figure 1.

PMLOSM is composed of hard springs, stator, and mover. The stator consists of coils, silicon steel lamination, stainless steel frame and stainless steel end covers. The mover consists of Halbach permanent magnetic array and iron frame. Springs are inserted between stator and mover, so the mover becomes a typical second order system which has a high inherent frequency. A diagram of the PMLOSM is depicted in Figure 2.

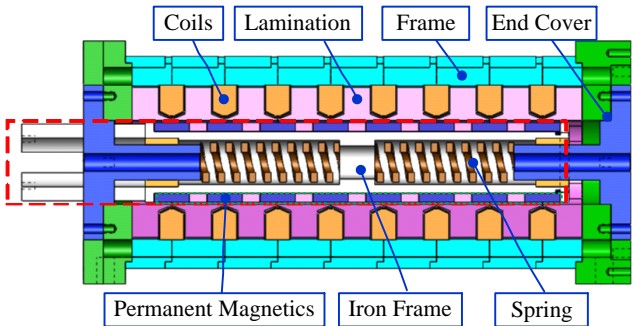

**Figure 1.** Cross section of PMLOSM.

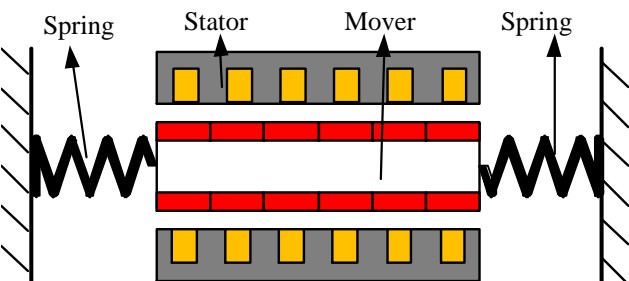

**Figure 2.** PMLOSM diagram.

The insertion of hard springs makes up a typical second order system. If the input current is of inherent frequency, PMLOSM can operate at high frequency. The open loop magnitude figure is shown in Figure 3. If PMLOSM works at inherent frequency, the efficiency of PMLOSM would be high. However, the phase lag is about 90 degrees, as shown in Figure 4. When we design a closed loop controller, it is inevitable to conquer the phase lag to obtain high tracking precision, which challenges the controller design.

Hence, we proposed AMFC for PMLOSM to obtain high tracking precision. AMFC is a combination of feedforward control and traditional PI control. Traditional PI control satisfies stroke adjustment and feedforward control improves phase response.

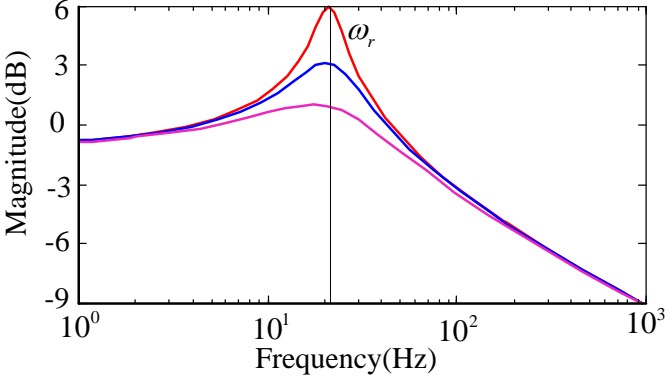

**Figure 3.** Magnitude response of PMLOSM.

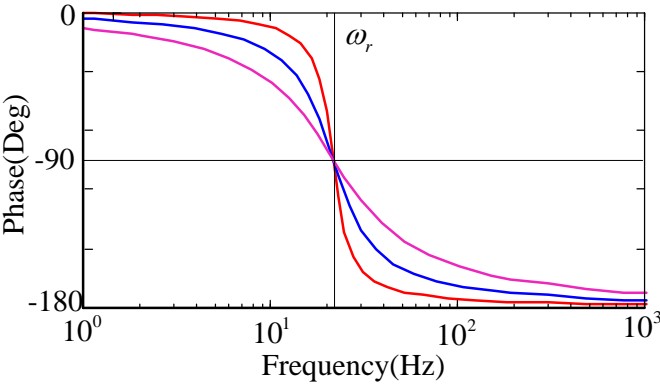

**Figure 4.** Phase response of PMLOSM.

## 3. Mathematical Model and Simulation Analysis

### 3.1. Mathematical Model

Assume that the PMLOSM is symmetrical and the relationship between thrust and input current is linear, absolutely obtained by experiment shown in Figure 5. Hence, the electromagnetic force of the PMLOSM can be written as

$$F_\mathrm{e} = K_\mathrm{e} i \tag{1}$$

where $F_\mathrm{e}$ is electromotive force of the PMLOSM; $K_\mathrm{e}$ is force constant coefficient of the PMLOSM; $i$ is input current.

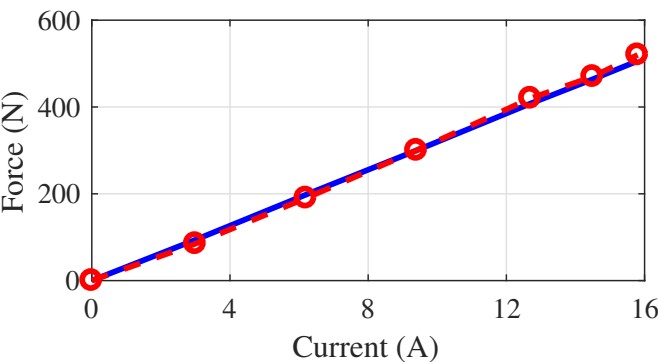

**Figure 5.** Relationship between input current and thrust.

Assume that mover is a rigid body and there is no deformation, and the dynamic function, based on Newton's second law, can be written as

$$F_\mathrm{e} = kx + \xi \dot{x} + m\ddot{x} \tag{2}$$

where $k$ is spring stiffness of PMLOSM; $m$ is total mass of mover and piston; $\xi$ is damping coefficient of LHP; $x$ is displacement of mover.

Traditional PI control applied to the PMLOSM is shown in Figure 6.

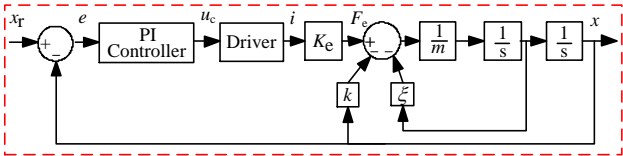

**Figure 6.** Traditional PI control sketch of PMLOSM.

The command error is written as

$$e = x_r - x \tag{3}$$

where $x_r$ is command signal.

Hence, the PI control law can be written as

$$u_c = K_i \int e dt + K_p e \tag{4}$$

where $K_i$ is integral gain; $K_p$ is proportional gain.

The relationship between input current and control law can be written as

$$i = K_d u_c \tag{5}$$

where $K_d$ is gain of current driver.

Substitute (4) into (5), and the input current can be written as

$$i = K_d K_i \int e dt + K_d K_p e \tag{6}$$

Substitute (6) into (1), and the electromagnetic force can be written as

$$F_e = K_e K_d K_i \int e dt + K_e K_d K_p e \tag{7}$$

Substitute (3) and (7) into (2), and the closed loop function can be written as

$$K_e K_d K_i \int (x_r - x) dt + K_e K_d K_p (x_r - x) = kx + \xi \dot{x} + m\ddot{x} \tag{8}$$

Perform Laplace transform at both sides of (8), and (8) can be transformed as

$$K_e K_d K_i \frac{1}{s}(X_r - X) + K_e K_d K_p (X_r - X) = kX + \xi s X + ms^2 X \tag{9}$$

So the closed loop system transfer function of traditional PI control can be expressed as

$$\frac{X}{X_r} = \frac{K_e K_d K_p s + K_e K_d K_i}{ms^3 + \xi s^2 + (k + K_e K_d K_p)s + K_e K_d K_i} \tag{10}$$

AMFC is combination of traditional PI control and feedforward control. The structure of AMFC is shown as Figure 7.

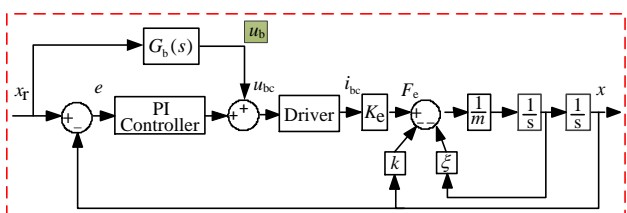

**Figure 7.** Combination of PI and feedforward control.

Control law of AMFC can be written as

$$u_{bc} = K_i \int e dt + K_p e + u_b \tag{11}$$

The driving current of PMLOSM can be written as

$$i_{bc} = K_d K_i \int e dt + K_d K_p e + K_d u_b \tag{12}$$

The dynamic equation of PMLOSM can be written as

$$K_e K_d K_i \int edt + K_e K_d K_p e + K_e K_d u_b = kx + \xi\dot{x} + m\ddot{x} \tag{13}$$

Substitute (3) into (13), and the dynamic equation can be written as

$$K_e K_d K_i \int (x_r - x)dt + K_e K_d K_p (x_r - x) + K_e K_d u_b = kx + \xi\dot{x} + m\ddot{x} \tag{14}$$

Perform Laplace transform at both sides of (14), and (14) can be transformed as

$$K_e K_d K_i \frac{1}{s}(X_r - X) + K_e K_d K_p (X_r - X) + K_e K_d U_b = kX + \xi sX + ms^2 X \tag{15}$$

where

$$U_b = X_r G_b(s) \tag{16}$$

Substitute (16) into (15), and the equation can be written as

$$K_e K_d K_i \frac{1}{s}(X_r - X) + K_e K_d K_p (X_r - X) + K_e K_d G_b(s)X_r = kX + \xi sX + ms^2 X \tag{17}$$

Hence the closed loop system transfer function of AMFC can be expressed as

$$\frac{X}{X_r} = \frac{K_e K_d K_i + K_e K_d K_p s + K_e K_d G_b(s)s}{K_e K_d K_i + (k + K_e K_d K_p)s + \xi s^2 + ms^3} \tag{18}$$

If excellent performance of PMLOSM is desired, Equation (18) should be

$$\frac{X}{X_r} = 1 \tag{19}$$

Hence the transfer function of feedforward control can be derived as

$$G_b(s) = \frac{U_b}{X_r} = \frac{k + \xi s + ms^2}{K_e K_d} \tag{20}$$

Perform inverse Laplace transform, Equation (20) can be expressed as

$$u_b = \frac{1}{K_e K_d}(k + \xi\dot{x}_r + m\ddot{x}_r) \tag{21}$$

Figure 8 depicts the features of (21).

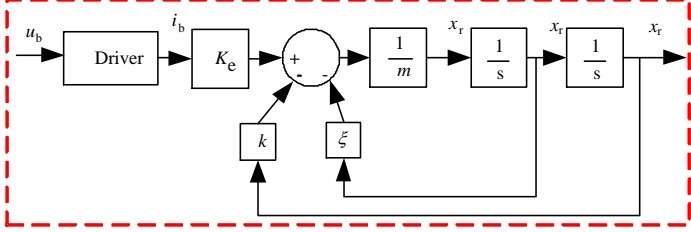

**Figure 8.** Features of (21).

Figure 8 shows that $x_r$ is the open loop response of $u_b$. If $x_r$ is

$$x_r = A\sin(2\pi ft + \varphi) \tag{22}$$

$u_b$ can be designed as

$$u_b = A_b \sin(2\pi ft + \varphi + \theta) \tag{23}$$

where $\theta$ is leading phase of $u_b$ with respect of $x_r$.

However, all analysis is linear model-based and the real physical model must be nonlinear. Hence the response of $u_b$ can be expressed as

$$K_d K_e u_b = k x_b + \xi \dot{x}_b + m \ddot{x}_b \tag{24}$$

$$x_b = \alpha x_r \tag{25}$$

where $\alpha$ is gain of AMFC.

The dynamic equation of PMLOSM can be written as

$$K_e K_d K_i \int (x_r - x) dt + K_e K_d K_p (x_r - x) + \alpha k x_r + \alpha \xi \dot{x}_r + \alpha m \ddot{x}_r = kx + \xi \dot{x} + m \ddot{x} \tag{26}$$

Perform Laplace transform at both sides of (26), and (26) can be transformed as

$$K_e K_d K_i \frac{1}{s}(X_r - X) + K_e K_d K_p (X_r - X) + \alpha k X_r + \alpha \xi s X_r + \alpha m s^2 X_r = kX + \xi s X + m s^2 X \tag{27}$$

Hence the system transfer function of AMFC can be expressed as

$$\frac{X}{X_r} = \frac{\alpha m s^3 + \alpha \xi s^2 + (K_e K_d K_p + \alpha k)s + K_e K_d K_i}{m s^3 + \xi s^2 + (K_e K_d K_p + k)s + K_e K_d K_i} \tag{28}$$

### 3.2. Simulation Analysis

The parameter values of PMLOSM are shown in Table 1.

**Table 1.** Parameters of PMLOSM.

| Symbol | Quantity | Value |
|--------|----------|-------|
| $K_e$ | Force constant | 32 N/A |
| k | Spring stiffness | 30,700 N/m |
| $\xi$ | Damping coefficient | 60 N s/m |
| m | Total mass | 1.35 kg |

If $\alpha = 0.5$, comparisons of amplitude and phase between PI control and AMFC are depicted in Figures 9 and 10, respectively.

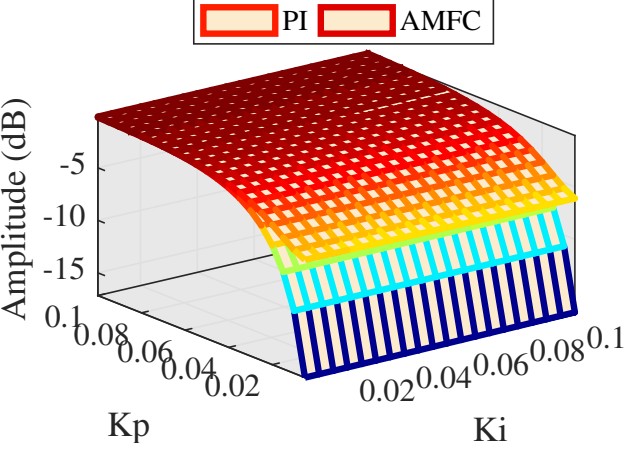

**Figure 9.** Amplitude comparison between PI and AMFC $\alpha = 0.5$.

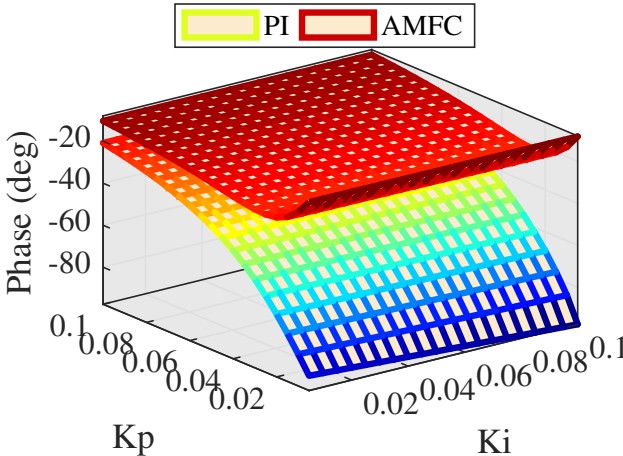

**Figure 10.** Phase comparison between PI and AMFC $\alpha = 0.5$.

As can be seen Figure 9, the amplitude response of AMFC performs a little better than PI control at low PI parameters, and Figure 10 shows that AMFC performs much better phase response than PI control and the phase lag is reduced from 20 to 10 degrees. This indicates AMFC improves phase response and it is necessary to enhance value of $\alpha$.

If the value of $\alpha$ increases up to 1, comparisons of amplitude and phase between PI control and AMFC are depicted in Figures 11 and 12, respectively.

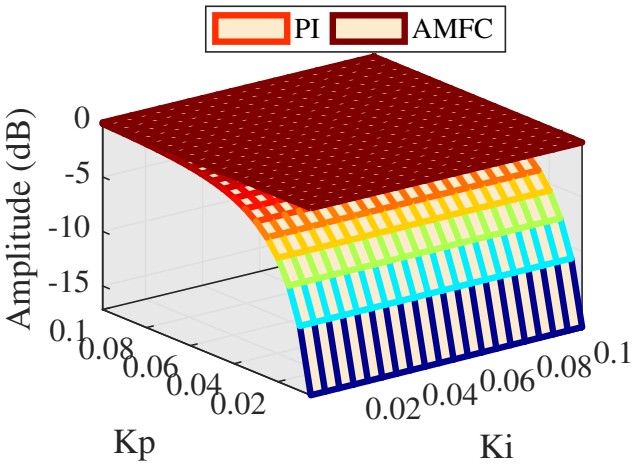

**Figure 11.** Amplitude comparison between PI and AMFC $\alpha = 1$.

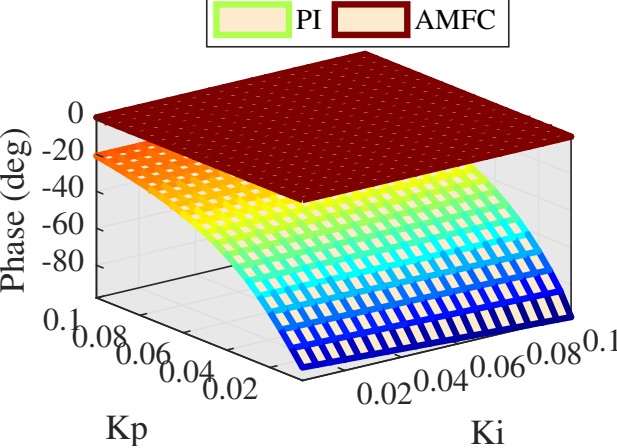

**Figure 12.** Phase comparison between PI and AMFC $\alpha = 1$.

As can be seen Figures 11 and 12, the amplitude response is 0 dB and phase lag is 0 degree invariably, showing extremely excellent performance of tracking accuracy. While it is suspected to design a motivation control signal that $\alpha = 1$ exactly. Hence it is inevitable to see the result when the value of $\alpha$ is higher than 1.

If value of $\alpha$ increases up to 1.5, comparisons of amplitude and phase between PI control and AMFC are depicted in Figures 13 and 14, respectively.

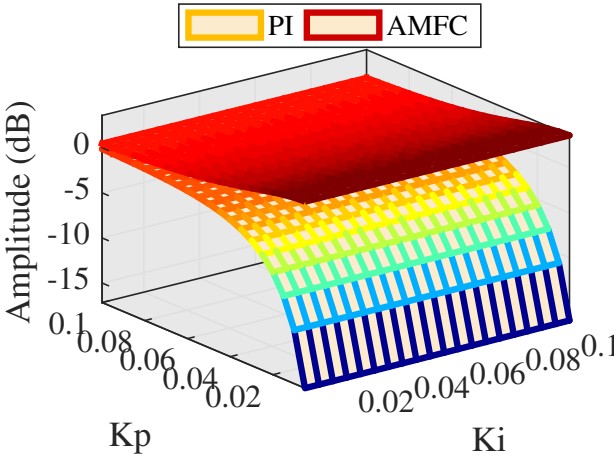

**Figure 13.** Amplitude comparison between PI and AMFC $\alpha = 1.5$.

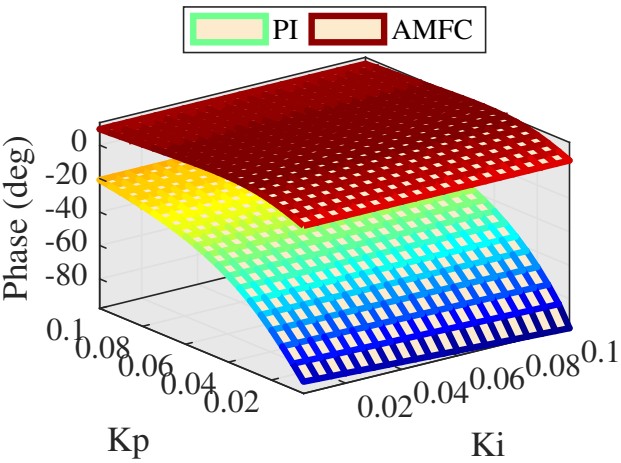

**Figure 14.** Phase comparison between PI and AMFC $\alpha = 1.5$.

As can be seen in Figure 13, amplitude response performs from 0.45 to 3.5 dB, and amplitude response decreased as the proportional gain increases, indicating that PI control plays the role of restraining the amplitude response. Figure 14 depicts the phase response to be an about 10 degrees advance, indicating that AMFC leads to an advancing system of loose tracking performance at $\alpha = 1.5$.

The above simulation research shows that the performance of PMLOSM varies with the changing of motivation gain and if $\alpha = 1$, the performance of PMLOSM is of high tracking precision. For this all, it is of great importance to search effects of motivation gain $\alpha$ surrounding 1 on the tracking performance. Motivation gain $\alpha$ is set from 0.8 to 1.2 and simulation results are shown in Figures 15 and 16.

As can be seen in Figures 15 and 16, amplitude and phase response tend to perform better as the motivation gain $\alpha$ increases, except if $\alpha = 0.9$. If $\alpha = 0.9$, amplitude and phase response mainly vary with proportional gain parameter. If $\alpha = 0.8$, phase response performs well and has small phase lag from $-1.2$ to $-2.6$ degrees but the amplitude response attenuates from $-1.65$ to $-0.48$ dB. If $\alpha = 1.1$ or $\alpha = 1.2$, the phase response

performs well, having small phase lag but amplitude has a large overshot that may cause system disorder. Hence, motivation gain $\alpha$ should be set close to 1 and simulation results are shown in Figures 17 and 18.

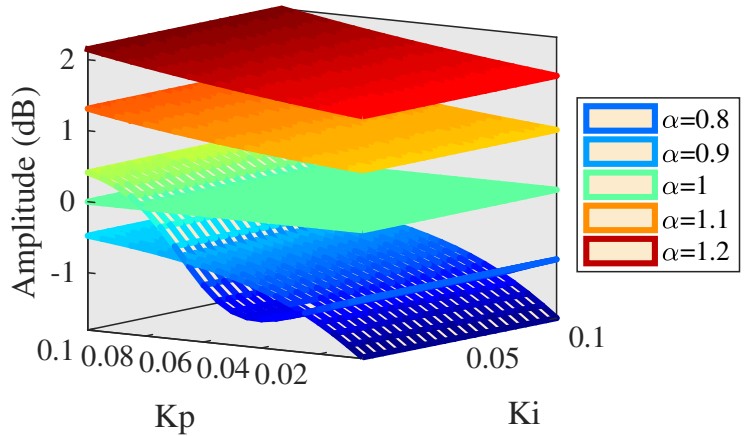

**Figure 15.** Amplitude of AMFC ($\alpha$ = 0.8 to 1.2).

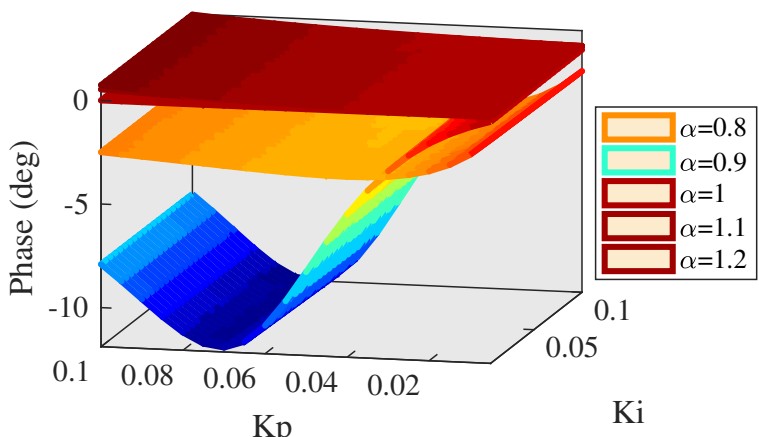

**Figure 16.** Phase of AMFC ($\alpha$ = 0.8 to 1.2).

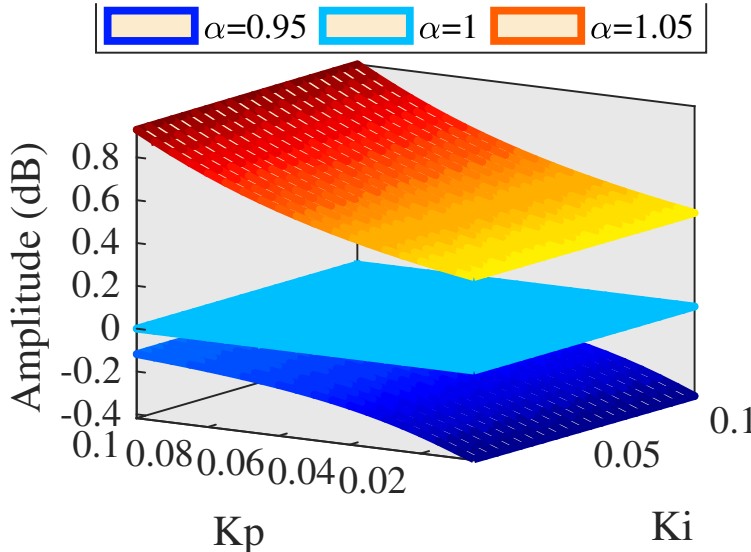

**Figure 17.** Amplitude of AMFC ($\alpha$ = 0.95 to 1.05).

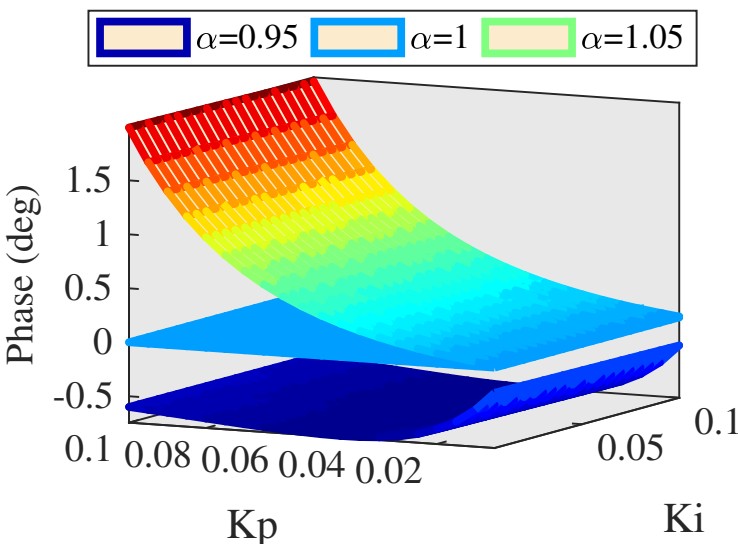

**Figure 18.** Phase of AMFC ($\alpha$ = 0.95 to 1.05).

As can be seen in Figures 17 and 18, both amplitude and phase response can perform well with accurate tracking performance and small fluctuation.

The above simulation results indicate that AMFC can provide high tracking accuracy for PMLOSM and perform much better than single traditional PI control. Moreover, motivation gain $\alpha$ plays a crucial role of tracking accuracy for PMLOSM, and motivation gain $\alpha$ should be set from 0.95 to 1.05 so that both amplitude and phase response can satisfy high tracking precision. Moreover, this indicates that feedforward controller has enough tolerance for experiment verification, and that even though the theoretical model has some error compared with the physical model, AMFC can provide high tracking accuracy for PMLOSM.

## 4. Experiment Results

The prototype of the PMLOSM and signal acquisition and control system are shown in Figures 19 and 20.

At the beginning of the LHP test, parameters of PI controller are set at a low level. Proportional gain is set from 0.03 to 0.04 and integral gain is set from 0.02 to 0.05. Amplitude and phase at low PI parameters are shown in Figures 21 and 22.

As can be seen in Figure 21, low PI parameters can reach a reliable amplitude response at high frequency, while Figure 22 shows that low PI parameters cannot achieve accurate phase tracking. The proportional parameter has more influence on phase response compared with that of integral parameter. Phase lag decreases as the proportional parameter becomes larger. Next step, proportional gain is set from 0.04 to 0.07 and integral gain is set from 0.03 to 0.05, and the experiments of larger proportional parameters are described in Figures 23 and 24.

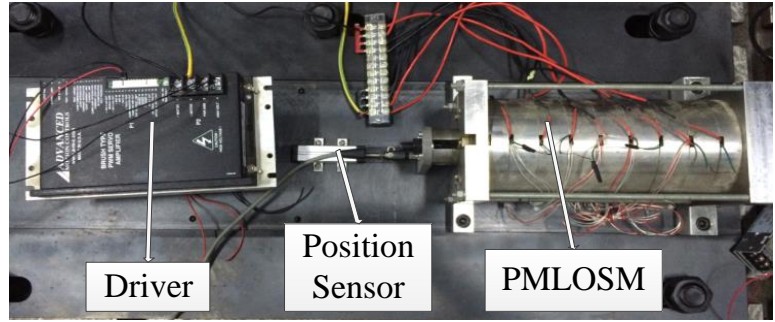

**Figure 19.** PMLOSM prototype.

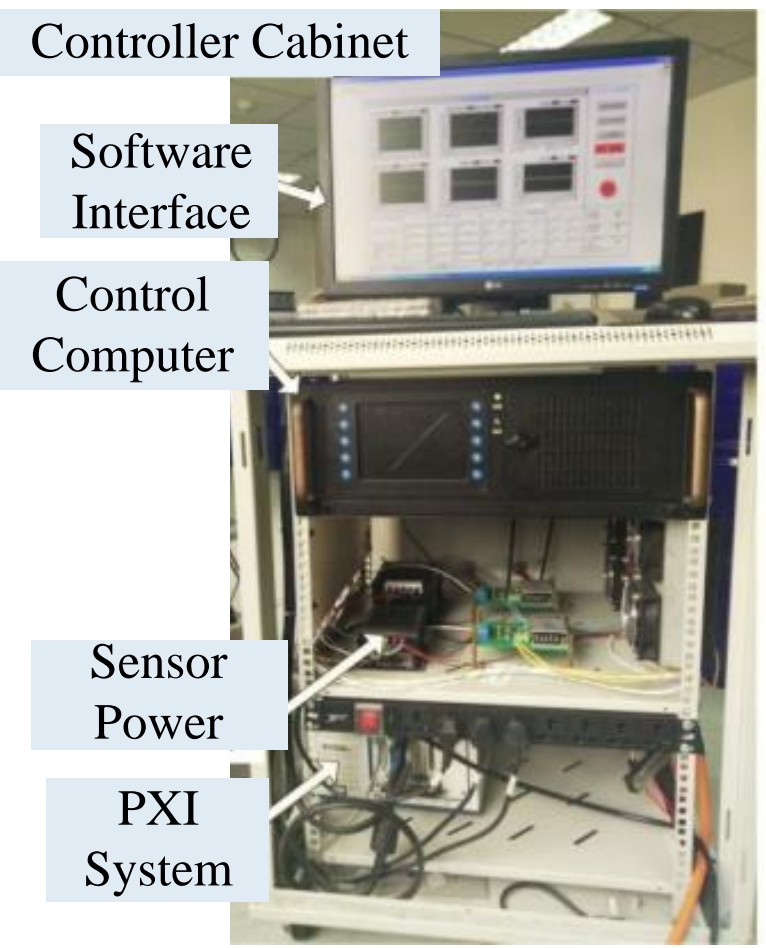

**Figure 20.** Signal acquisition and control system.

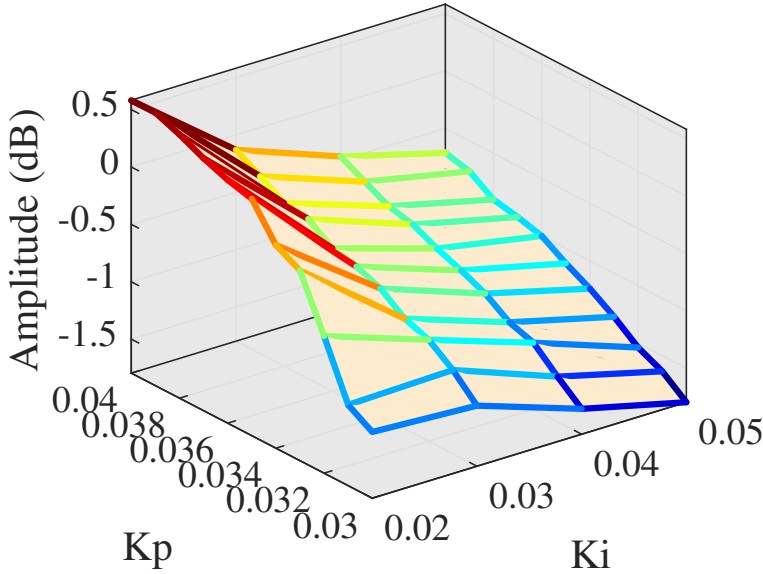

**Figure 21.** Amplitude at low PI.

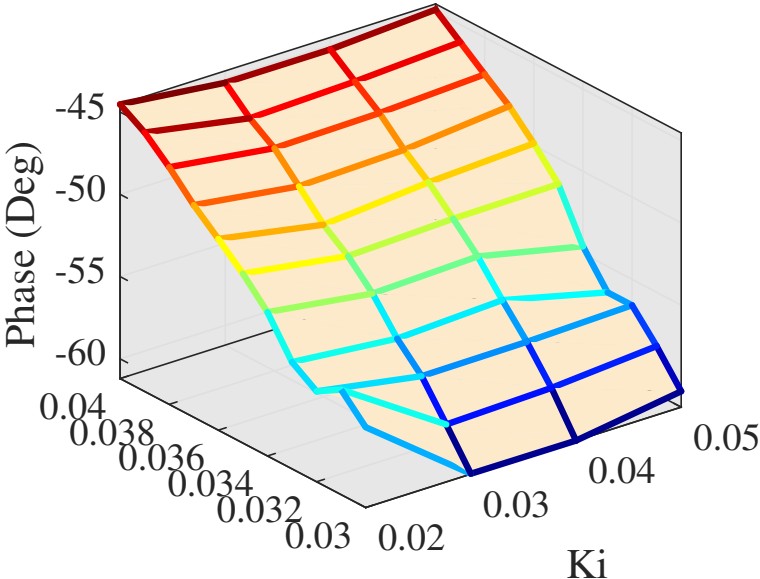

**Figure 22.** Phase lag at low level PI.

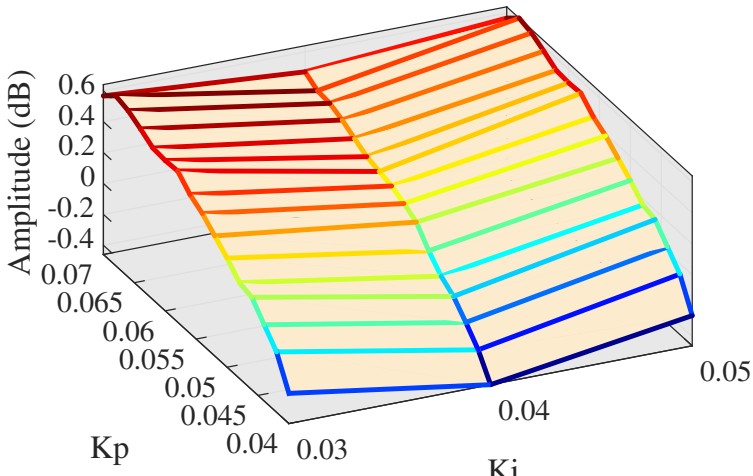

**Figure 23.** Amplitude at middle level PI.

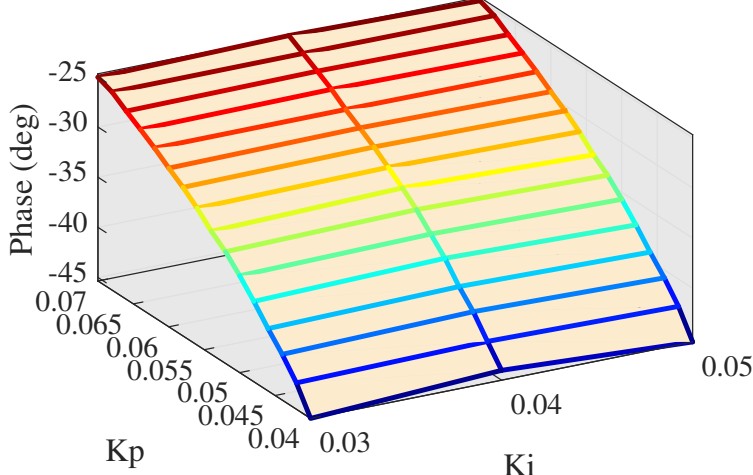

**Figure 24.** Phase lag at middle level PI.

As can be seen in Figure 23, middle PI parameters can also satisfy the requirement of amplitude response. However, if the proportional gain continues to increase, the amplitude response tends to be divergent. Moreover, Figure 24 shows that phase lag is at a high level though, decreasing from 45 to 25 degrees as the proportional gain increases. Phase response cannot satisfy the needs of high-accuracy tracking. Hence, it is necessary to increase both proportional gain and integral gain. Then, proportional gain is set from 0.07 to 0.1 and integral gain is set from 0.06 to 0.1. Experiment result of high level parameters is depicted in Figures 25 and 26.

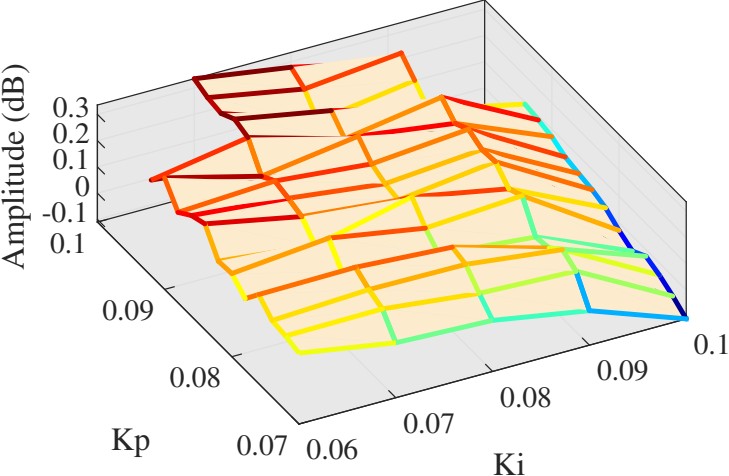

**Figure 25.** Amplitude at high level PI.

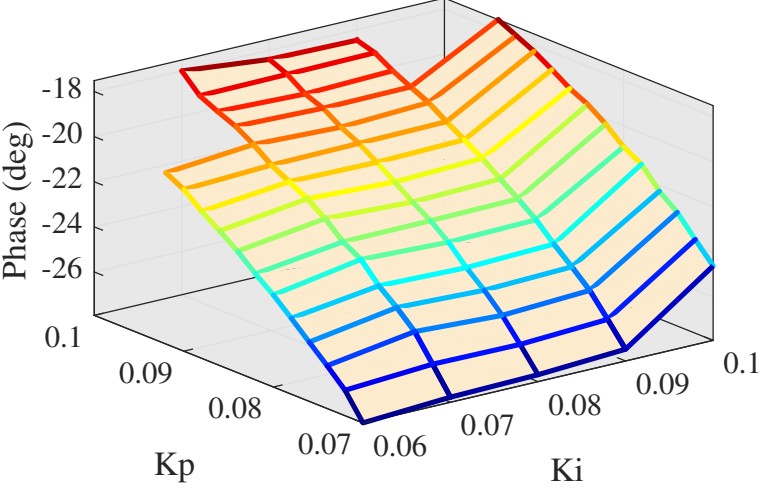

**Figure 26.** Phase lag at high level PI.

As can be seen in Figure 25, PMLOSM would appear to system crash at the high level PI parameters. Figure 26 shows that the best performance of phase response is −18 degrees at high level PI parameters, which cannot meet the high-accuracy tracking requirement. The reason why PI control cannot settle large phase lag is that the system stiffness is high and sample time of each trigonometric wave is large. Hence, we proposed a novel method called AMFC to satisfy the requirement of high-accuracy tracking of high frequency PMLOSM.

Firstly, the PI parameters of AMFC are set as $K_i = 0.04$ and $K_P = 0.02$ and the motivation gain $\alpha$ is from 0.5 to 1. The experiment result is show in Figures 27 and 28.

As can be seen in Figures 27 and 28, both amplitude response and phase response reach for better performance as motivation gain $\alpha$ tends to be 1. Figure 27 shows that motivation gain should be larger than 0.5 so that amplitude response is larger than −3 dB.

Moreover, if the motivation gain is larger than 0.8, system overshoot occurs. Figure 28 shows that the higher the motivation gain, the smaller the phase lags. If motivation gain is 1, the phase lag is −2.7 degrees. The position output is shown in Figure 29.

In [35], Kim studied the PID controller on an oscillating linear motor. The experiment results of PID control are shown from Figure 30. An experiment phase lag comparison is shown in Table 2 with respect to AMFC, PID, and PI.

AMFC obviously provided better dynamic performance than PI and PID.

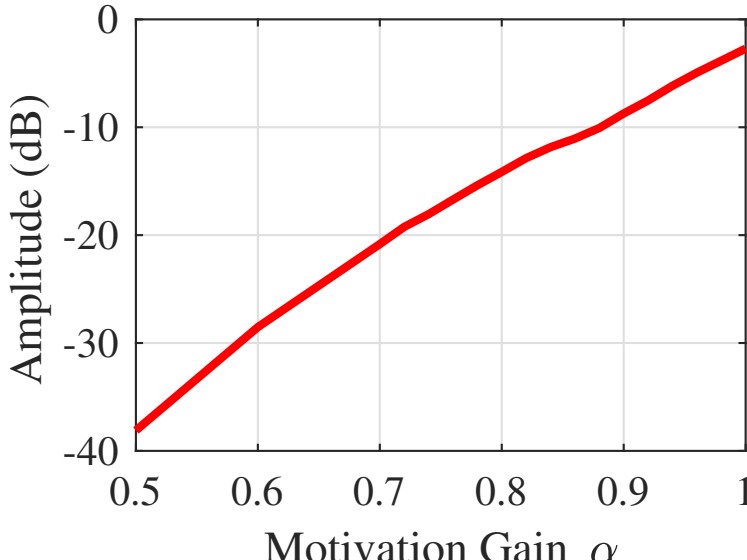

**Figure 27.** Amplitude of AMFC $\alpha$ = 0.5 to 1.

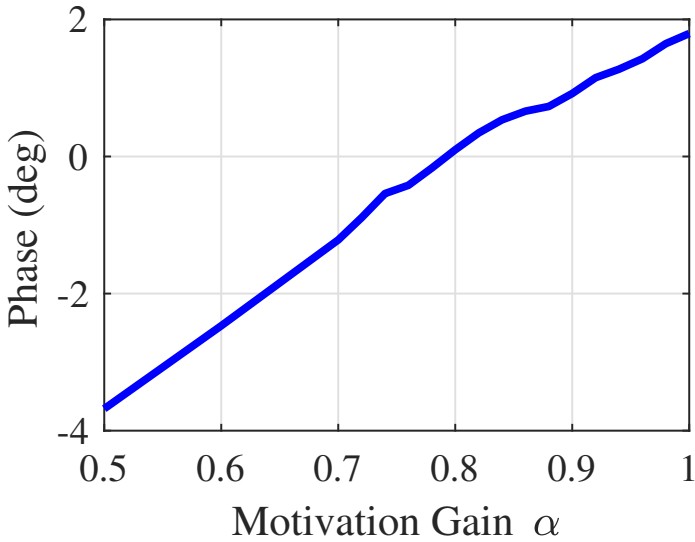

**Figure 28.** Phase of AMFC $\alpha$ = 0.5 to 1.

**Table 2.** Experiment phase lag of PI, PID and AMFC.

| Control Method | Phase Lag/deg |
|:---:|:---:|
| AMFC | 2.7 |
| PID | 6.4 |
| PI | 18 |

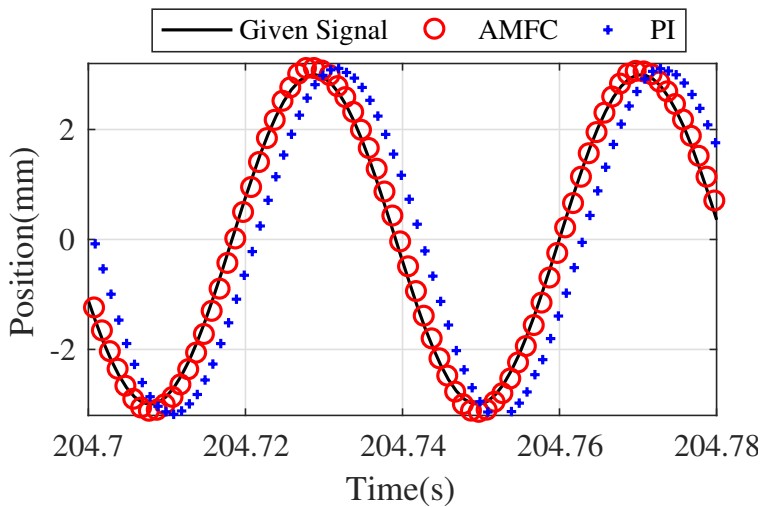

**Figure 29.** Position output of PMLOSM of PI and AMFC.

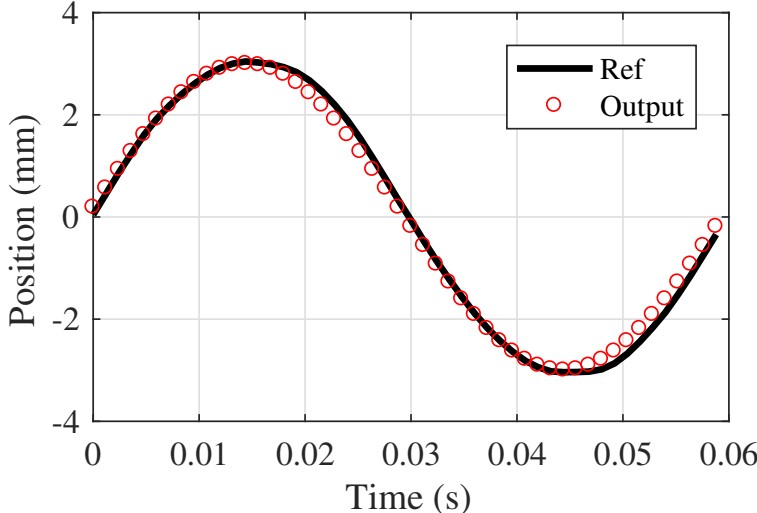

**Figure 30.** Experiment of PID.

The working frequency is set as 24 Hz and waveforms of current and position for one operating point of AMFC are shown in Figures 31 and 32.

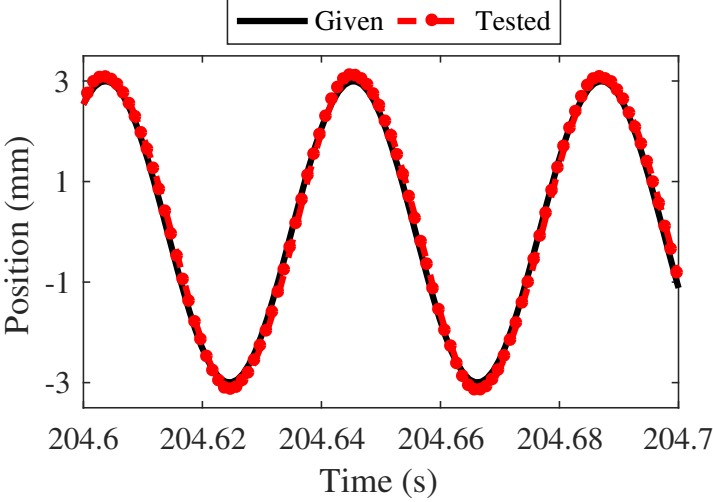

**Figure 31.** Position of AMFC.

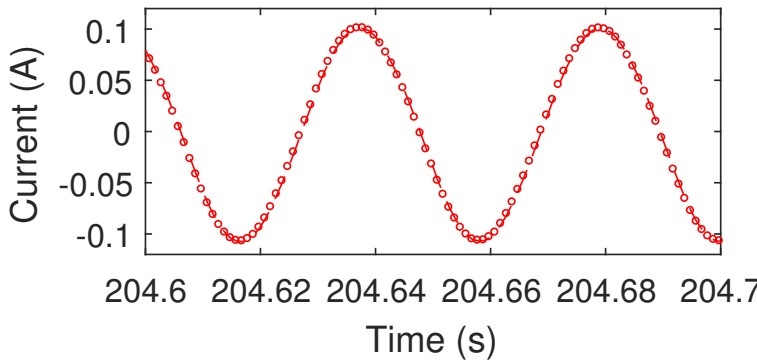

**Figure 32.** Drive current of AMFC.

Figure 31 shows that AMFC can provide highly accurate position tracking for PM-LOSM at high working frequency. The frequency characteristics due to different gain are in Figures 33 and 34.

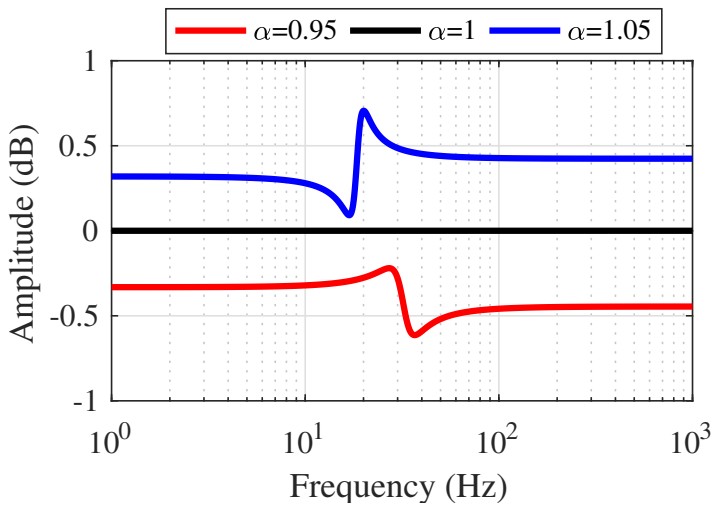

**Figure 33.** Amplitude–frequency characteristics of AMFC.

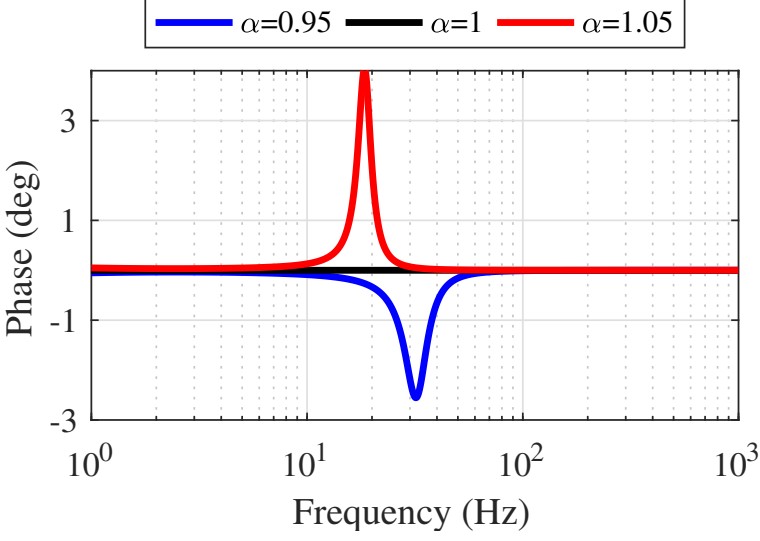

**Figure 34.** Phase–frequency characteristics of AMFC.

As can be seen in Figures 33 and 34, AMFC can provide accurate position tracking at high working frequency for PMLOSM. If a small unknown force load acts on the PMLOSM, the position output is shown in Figure 35.

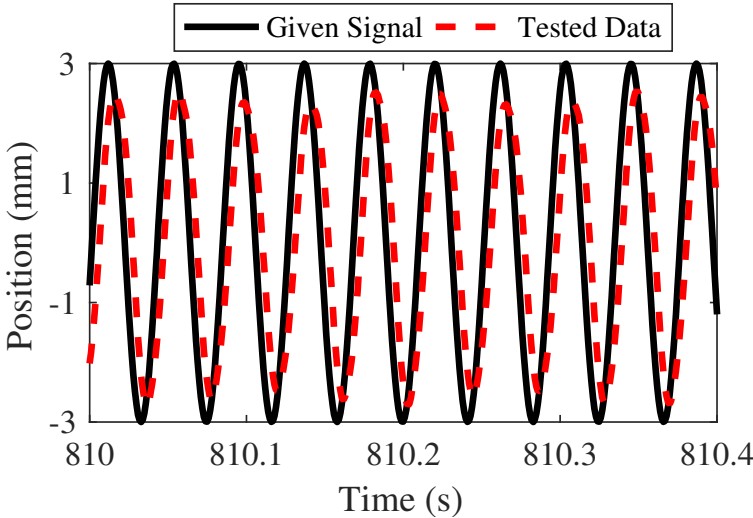

**Figure 35.** Position with load.

Figure 35 shows that AMFC has a certain degree of robustness.

## 5. Conclusions

Linear motors can provide direct motion without any other transmission gears or power producers such as fluid pumps and air compressors. A permanent magnetic linear oscillating synchronous motor is composed of a permanent magnetic linear synchronous motor and two springs to construct a typical second order mass-spring system, so that the permanent magnetic linear oscillating synchronous motor can provide high frequency reciprocating motion, which is challenging for controller design because of limited sampling time, high spring stiffness, and high motor operating frequency.

In this paper, detailed structure illustration and problem formulation are provided, and it is a great challenge to guarantee high tracking precision of PMLOSM because of large phase delay due to hard springs and high operating frequency. The proposed AMFC is a combination of feedforward control and traditional PI control. The simulation results show that AMFC performs better than single traditional PID control and motivation gain should be set from 0.95 to 1.05 to show that the feedforward controller has enough tolerance, which indicates that even though the theoretical model has some error compared with the physical model, AMFC can provide high tracking accuracy for PMLOSM. The experiment results show that AMFC decreases the phase lag from −18 to −2.7 degrees and maintains enough amplitude response, and AMFC can provide high tracking precision for PMLOSM at a high frequency. In conclusion, AMFC gets the hang of how to achieve high tracing precision of PMLOSM.

More illustrations: In this paper, we mainly focus on the tracking performance of PMLOSM differential control and limited robustness belongs to the motor. In addition, system analysis is based on linear model system without taking load nonlinearity into consideration. In the future, we will concentrate on load nonlinearities such as instantaneous load, constant large load, and nonlinear friction to improve the robustness of the PMLOSM.

**Author Contributions:** Conceptualization, Z.J., Y.C. and L.Z.; Funding acquisition, Z.J.; Project administration, Z.J.; Data curation, Y.C.; Formal analysis, Y.C. and L.Z.; Methodology, Y.C.; Resources, Y.C. and Y.L.; Validation, Z.J.; Software, X.L. and Y.L.; Investigation, L.Y.; Supervision, L.Y.; Writing—Original draft, Y.C.; Writing—Review and editing: Y.C., X.L., L.Z. and Y.L. All authors have read and agreed to the published version of the manuscript.

**Funding:** This work was supported by, the National Key R&D Program of China under grant 2017YFB1300400 the National Natural Science Foundation of China under grants 51890882, 51875013 and 51575026, National Key Basic Research Program of China under grant 2014CB046400, the New Generation of Artificial Intelligence under grant 2018AAA0102900.

**Institutional Review Board Statement:** The study was conducted according to the guidelines of Science and Technology on Aircraft Control Laboratory, Beihang University, and approved by School of Automation Science and Electrical Engineering, Beihang University.

**Informed Consent Statement:** Informed consent was obtained from all subjects involved in the study.

**Data Availability Statement:** The data that support the findings of this study are available from the corresponding author upon reasonable request.

**Acknowledgments:** This work was supported by, the National Key R&D Program of China under grant 2017YFB1300400 the National Natural Science Foundation of China under grants 51890882, 51875013 and 51575026, National Key Basic Research Program of China under grant 2014CB046400, the New Generation of Artificial Intelligence under grant 2018AAA0102900, the Fundamental Research Funds for the Central Universities, Science and Technology on Aircraft Control Laboratory, and Ningbo Institute of Technology, Beihang University.

**Conflicts of Interest:** The authors declare no conflict of interest.

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
