# Peer review of "Advancing Motivation Feedforward Control of Permanent Magnetic Linear Oscillating Synchronous Motor for High Tracking Precision"

_actuators, doi:10.3390/act10060128_

Round 1

Reviewer 1 Report

Dear authors,

I appreciate the work you have done in this research. I have few questions and notes that should be clarified in the paper.

How can you simplify your motor to such simple equation as (1)? What do you mean by extremely symmetry? What all did you neglect? The resistances, the inductances, the magnetic saturation, the windings, the sizes of magnets, cogging force, EMF, load angle?

How can you derive equation (5)? What driver are you referring to? Is it a current source controlled by voltage?

Could you show whole control structure with schematics of power components connection and with sensors necessary for operation?

Which sensors were used? I see only position sensor, but I assume that some other variables were also measured.

Could you show the waveforms of current, speed, force, and position for one operating point?

Provide more information about the experiment like the motor rated voltage, motor rated power, used controller, sampling time, discretization type of the control algorithm, switching frequency, driver, ...

Does the load affect the performance?

Equation (21) shows the design of the feedforward part of the proposed control. The calculation involves double integrations, which may suffer from offset and noise. How is the integration made in the controller? Further, the knowledge of parameters Kd, Ke, k, ξ, and m, which are usually not exactly known and may change over time and over other parameters, is assumed. With what precision must they be known for reliable operation? When they are not known with sufficient accuracy, couldn’t this feedforward part make more harm than good?

In Figs. 3 and 4, what do the different colors show?

Author Response

The authors would like to sincerely appreciate your constructive comments on our paper, “Advancing Motivation Feedforward Control of Permanent Magnetic Linear Oscillating Synchronous Motor for High Tracking Precision”. Comments will help us to have a broader view of research issues as well as to further consolidate the paper.

Following the valuable suggestions, we will carefully check through the manuscript, and make proper revisions accordingly.

Sincerely appreciate your kind assistance. Thank you very much!

Reviewer 2 Report

This paper proposed an AMFC based on the traditional PI controller to obtain high tracking precision of PMLOSM.

This work takes a serial of experiments and show motion performance, and the law of motion profiles related to different control parameter is concluded. In my opinion, the control method is a type of general feedforward compensator. It is predictable that the phase characteristics with the feedforward compensator can be improved. Here are my comments.

1 I think there are two challenges in this work. Firstly, we can not get the accurate model parameter which is shown in Table 1. Secondly, it is hard to get the ideal differentiator in the system design. If the two challenges can be overcome, the system can work without the phase difference in theory. I think the authors would consider the two issues in their work.

2 In the test, the authors undertake plenty of experiment to summarize the control rule, but more theoretical contribution should be presented, or it lacks the guideline for other motion control system.

3 The frequency of the sine trajectory is constant. The frequency characteristics due to different gain a would make this work more cogent.

Author Response

(The authors gave the same response as above.)

Reviewer 3 Report

In this paper, the authors have proposed an advancing motivation feedforward control (AMFC), combination of advancing motivation signal and PI control signal to obtain high tracking precision of PMLOSM. The study is very interesting. However, several concerns in the current stage of manuscript need to be addressed before consideration of acceptance, as follows:

To make the manuscript self-contained and more readable, make sure all the variables have been fully specified after each equation.

Please add more discussion and limitations of the proposed method.

More discussions and literatures should be added in the introduction.

Is there any other limitation of the proposed idea? Such as the specific or limited ranges of working environment or conditions? Is the proposed architecture still valid under other environment?

Carefully recheck grammar and typo errors.

Author Response

(The authors gave the same response as above.)

Round 2

Reviewer 1 Report

All points have been corrected.

Reviewer 2 Report

Your responses make sense.